# Assembly of the salt-secreting mangrove *Avicennia rumphiana*

Jeremy R. Shearman[1], Chaiwat Naktang[1], Chutima Sonthirod[1],
Wasitthee Kongkachana[1], Sonicha U-thoomporn[1], Nukoon Jomchai[1],
Chatree Maknual[2], Suchart Yamprasai[2], Poonsri Wanthongchai[2], Wirulda Pootakham[1],
Sithichoke Tangphatsornruang[1]*

1 National Center for Genetic Engineering and Biotechnology (BIOTEC), National Science and Technology Development Agency (NSTDA), Pathum Thani, Thailand, 2 Department of Marine and Coastal Resources, 120 The Government Complex, Thung Song Hong, Bangkok, Thailand

* sithichoke.tan@nstda.or.th

## Abstract

*Avicennia rumphiana*, also known as *Avicennia marina* var. *rumphiana* or *Avicennia lanata*, is a mangrove species that has high salt tolerance and is one of the few species that secretes salt through the leaves, similar to *Avicennia marina*. We sequenced and assembled the *A. rumphiana* genome into 24,094 Supernova-scaffolds totalling 499.6 Mb. Sequence comparison showed that 68.7% of the *A. rumphiana* genome aligned to *A. marina* sequence, covering 72% of the *A. marina* genome at an average nucleotide identity of 87.7%, showing that *A. marina* is closely related to A. rumphiana and, thus, suitable for reference based scaffolding of the *A. rumphiana* Supernova-scaffolds. Reference based scaffolding produced 32 chromosome-level scaffolds containing 447.3 Mb, with 52.3 Mb of sequence unplaced. Annotation of this genome assembly resulted in 37,347 genes with 22,414 of those contained within chromosome scaffolds. A total of 671 genes matched to genes that are linked to salt tolerance. Genome comparison shows that *A. rumphiana* is quite different to *A. marina* and should, therefore, not be referred to as a variant of *A. marina*.

## Introduction

*Avicennia rumphiana* is a mangrove species with high salt tolerance that secretes salt through the leaves, similar to *Avicennia marina*. It is, perhaps, the similarities to *Avicennia marina* that are responsible for the history of controversial taxonomic classification of this species, a controversy that still exists. The species was first reported as *Avicennia rumphiana Hallier f* in 1918, which pre-dates the naming as *Avicennia lanata Ridley* in 1920 by two years. To add further confusion, this species was named *A. marina* var. *rumphiana (Hallier f.) Bakh* in 1921. All of these classifications were reviewed by Duke [1] and determined to reference the same species, which led to the conclusion that the species should, therefore, be named *Avicennia rumphiana*, because this name has precedent. However, the situation is further obfuscated by publication of the species as *Avicennia lanata Ridley* in Tomlinson ([2] and updated in [3]) with reference of it previously being named *A. officinalis* L. var. *spathulata Kuntze* in

**Data availability statement:** The genome assembly, raw reads, and population RADseq for this work can be found at NCBI under the bioproject PRJNA797928.

**Funding:** This work was funded by the National Science and Technology Development Agency, Thailand. The funders had no role in study design, data collection and analysis, decision to publish, or preparation of the manuscript.

**Competing interests:** The authors declare that there are no competing interests.

1891, which itself is considered a synonym for *Avicennia marina* var. *rumphiana*. Thus, all of these names appear to reference the same species and various databases use either *Avicennia lanata* or *Avicennia marina* var. *rumphiana*. The publication record for this species, as a result, is complicated with many publications using the name *A. lanata*, some using the name *A. rumphiana*, and others using the name *A. marina* var. *rumphiana*, often with insufficient awareness that all are referencing the same species.

*Avicennia rumphiana* is one of the few species that secrete salt onto the leaf surface via salt secreting glands [4]. These glands result in salt visibly crystallizing on the surface of the leaves. Such a mechanism has several potential applications if it can be suitably utilised, ranging from adding salt tolerance to existing crop species to the potential for a plant that can filter and remove salt from highly saline soil. An effective way to understand salt excretion is to compare salt excretion genes between each of the species that use this mechanism. One salt secreting species with several high quality reference genomes available is *Avicennia marina* [5–7]. Considering only species with a chromosome-level genome assembly available, *A. marina* is the most closely related species to *A. rumphiana*. The genome of *A. marina* was analysed to identify genes in the salinome (salinity response genes) and represents an important reference for comparison of species that excrete salt through the leaves [5,7].

A high quality reference genome was published for *A. rumphiana* as part of a study that sequenced and assembled 48 species [8]. That study used long PacBio reads to achieve an N50 of 7Mb, but did not include any scaffolding step, such as Hi-C. So, while the *A. rumphiana* genome (GWHBCJH00000000) was already published, it was analysed as part of a large group with limited species specific analysis [8]. It has been observed that a single reference genome captures approximately 70-90% of the unique sequence that exists within the entire species [9], and that some of this variation can included presence/absence alleles for entire genes. For example, a pan-genome of *Brachypodium distachyon* contained approximately twice the number of genes found in any single individual [10]. As a result, there is now a push toward developing what is known as the pan-genome reference, which is an assembly construct that incorporates the sequence and genomic features of a large number of individuals from the species. Thus, having multiple assemblies for each species is necessary to fully capture the variation that exists within each species, not only for non-coding sequence, but for actual genes also. Here we present another *A. rumphiana* genome assembly, annotation, and comparison of the salinity-response genes to *A. marina*. Furthermore, we recommend sole use of the name *A. rumphiana* for all future publications regarding this species.

## Materials and methods

### Whole genome sequencing and assembly

Fresh young leaf was collected from a single *A. rumphiana* tree from the Ranong Biosphere Reserve, Thailand, and placed into liquid nitrogen. All necessary permissions were obtained for the sample collection. Frozen leaf was used for DNA extraction with the standard CTAB method followed by clean-up using a DNeasy Mini spin column from Qiagen.

DNA was used to prepare a linked-read library [11,12] using 10x genomics and sequenced on an Illumina HiSeq following Illumina protocols for 150 bp paired-end reads. De novo genome assembly was performed using supernova v2.1.1 [13] with default settings and the 'pseudohap output' style using an estimated genome size based on the *A. marina* genome (JACDXK000000000). Supernova includes a scaffolding step making use of the linked read data, so the output is referred to as 'Supernova-scaffolds' instead of 'contigs'. The genome assembly was checked using Merqury v1.3 [14] with all raw reads and the Supernova-scaffolds allowing the program to calculate the optimum k-mer size. The genome was scaffolded using

RagTag v2.0.1 [15] with *A. marina* (JACDXK000000000) as the reference to generate chromosome level scaffolds and these are referred to as 'RagTag-scaffolds' to avoid confusion with the output from Supernova. The default settings for RagTag were used with the instruction to not split contigs (Supernova-scaffolds) for cases where contigs (Supernova-scaffolds) had split mapping results. The reliability of this RagTag-scaffolding step was investigated by mapping the sequence of the RagTag-scaffold assembly against *A. marina* to identify any RagTag-scaffolds with discontiguous mapping. In addition, the published *A. rumphiana* genome (GWHBCJH00000000) [8] was scaffolded to *A. marina* using the same RagTag method to identify any cases of discontiguity.

## Annotation

Annotation was performed for *A. rumphiana* on the repeat-masked genome using EvidenceModeler v1.1.1 [16] using gene sets and protein sets from related species and *ab initio* gene prediction. Protein sequences were obtained from public databases for *Avicennia marina*, *Oryza sativa*, *Mimulus guttatus*, *Sesamum indicum*, *Populus trichocarpa* and *Eucalyptus grandis* and aligned to the genome using AAT [17]. The ab initio prediction program Augustus v3.3.3 was used to predict genes based on the genome assembly [16]. All gene predictions were then combined by EvidenceModeler to generate consensus gene models using equal weights for each evidence type. Genome completeness was estimated by comparing each annotation gene set against the embryophyta_odb10 data set using Benchmarking Universal Single-Copy Orthologs (BUSCO) v4.0.5 [18]. The predicted gene sets for each species were functionally annotated using OmicsBox v2.0.10 (https://www.biobam.com/omicsbox) [19].

The annotated proteins were blasted to the *A. marina* proteins to identify homologous genes. Putative salt genes were identified by blast match to known salt genes based on the list of known salt genes from Natarajan et al. [5]. A two sample t-test was performed on the percent identity match of the blast result with all known salt genes as the 'Salt' group and all genes excluding the known salt genes as the 'All' group using the 't.test' function in R.

## Repeat sequence annotation

De novo repeat families were identified and classified for *A. rumphiana* using RepeatModeler version 2.0.1 [20]. These repeat sequences were then aligned to Genbank's non-redundant protein database (using BLASTX with an E-value cutoff of $1 \times 10 - 6$) to exclude repeat sequences that contain large families of protein-coding genes.

## Comparative genetics

The Supernova-scaffolds of *A. rumphiana* were compared to *A. marina* using the nucmer program of MUMmer v3.23 [21] to identify collinearity within Supernova-scaffolds to estimate the suitability of *A. marina* for reference based scaffolding. Nucmer was also used for the *A. rumphiana* Supernova-scaffolds of our assembly and the published *A. rumphiana* assembly (GWHBCJH00000000). The nucmer outputs of whole-sequence alignment were plotted in R to identify rearrangements relative to *A. marina* to produce Fig 1 and S1 Fig. Genome mapping statistics were calculated using the CIGAR string from mapping the Supernova-scaffolds, RagTag-scaffolds, or contigs (GWHBCJH00000000) against the reference (either A. marina genome or A. rumphiana PacBio contigs (GWHBCJH00000000)) using BWA v0.7.18 mem [22]. The CIGAR string was parsed to extract start and end locations of the mapped sequences, dropping unmapped reads, with the percent identity calculated from the number of insertions and deletions. To avoid double counting any individual base, the resulting alignment-interval table was then reduced using the GenomicRanges 'reduce' function in R to

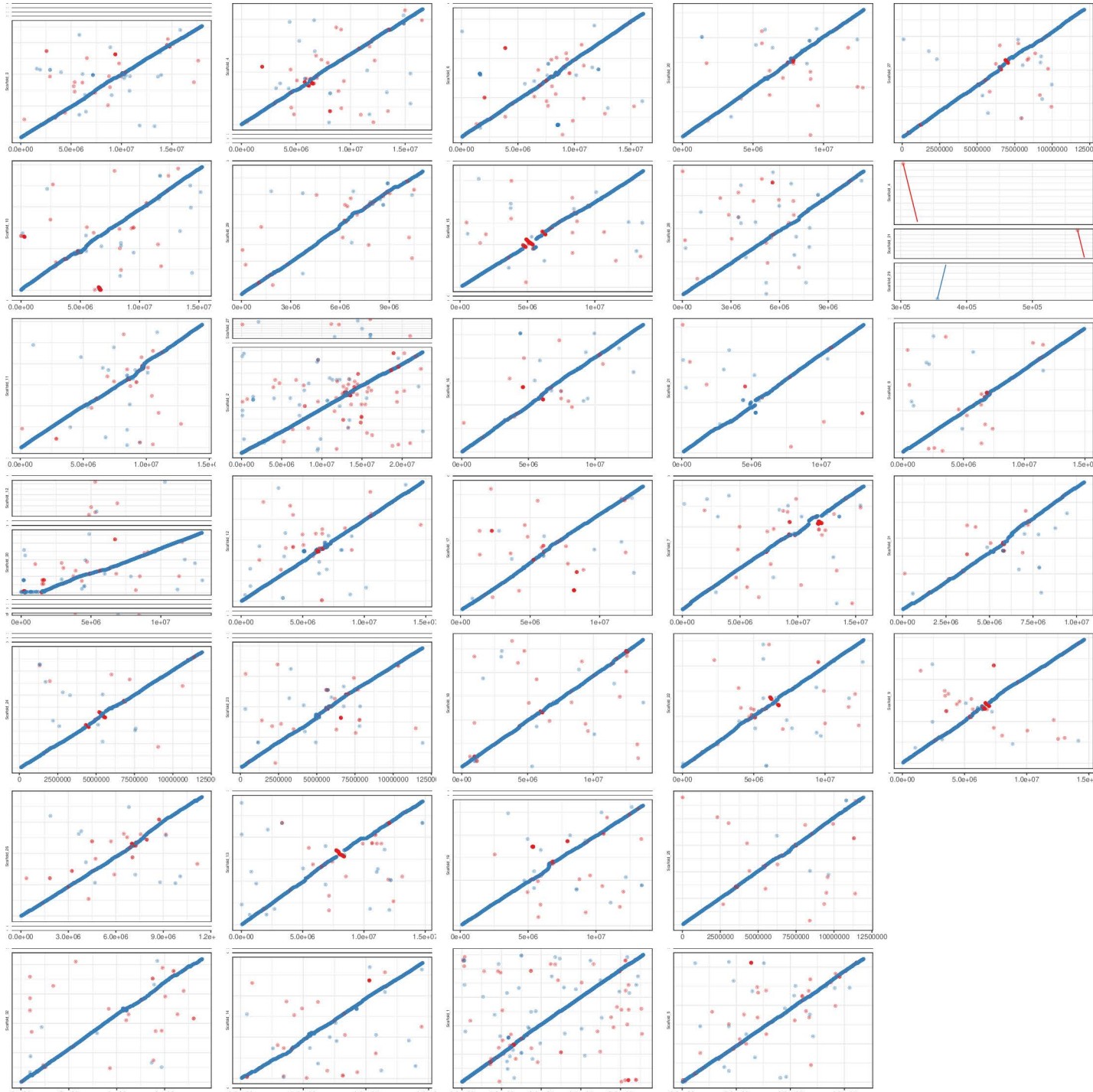

**Fig 1. *Avicennia rumphiana RagTag*-scaffold sequence mapped against *Avicennia marina* chromosome sequence.** The whole-genome sequence of the RagTag-scaffolds of A. *rumphiana* on the y-axes mapped against the whole-genome sequence of A. *marina* on the x-axes. Cases where multiple *A. rumphiana* RagTag-scaffolds map to a single *A. marina* chromosome appear as stacked boxes where the length of the y-axis is proportional to the amount of sequence that maps, resulting in small regions appearring as individual lines. Blue dots indicate direct sequence alignment, while red dots indicate inverted sequence alignment.

collapse overlapping intervals into a single interval. This reduction means that each base of the query sequence is counted only one time, even if it maps multiple times, allowing for calculation of the amount of query sequence that can map to the reference sequence. The reduce function loses the percent identity and location of each individual alignment, but the total percent identity of the mapped sequence can be estimated by taking the sum of each mapped sequence length multiplied by its percent identity divided by the sum of all mapped lengths.

We also mapped the raw 10x-sequencing reads against our genome assembly and the published genome assembly using BWA to report on the percent of raw reads that can map to each assembly.

The RAxML-NG program v1.0.2 [23] was used to construct a phylogenetic tree based on orthologous groups that were identified by OrthoFinder v2.5.5 [24]. The species included in the phylogenetic tree were *Arabidopsis thaliana*, *Avicennia marina*, *Bruguiera cylindrica*, *Bruguiera gymnorrhiza*, *Bruguiera parviflora*, *Bruguiera sexangula*, *Ceriops decandra*, *Ceriops tagal*, *Ceriops zippeliana*, *Kendelia obovata*, *Populus trichocarpa*, *Rhizophora apiculata*, *Ricinus communis*, and *Oryza sativa*. Protein sequences were aligned using MUSCLE [25] and a substitution model was estimated using the ModelTest-NG program v0.1.7 [26]. The divergence times in the phylogenetic tree were estimated by the Bayesian Relaxed Molecular Clock approach using MCMCtree v4.0 [27]. Divergence times were anchored using dated fossil records, the root node of the common ancestor of Rhizophoraceae, Euphorbiaceae (*R. communis*) and Salicaceae (*P. trichocarpa*) placed 105 - 120 MYA, fossils of the ancestor of Rhizophoreae from 48 - 56 MYA, and a fossil recognized as ancestral Rhizophora dated to 34 - 38 MYA. Colinear blocks were identified from the blastp alignment results using MCScanX [28] with regions of at least ten syntenic genes (with no more than six intervening genes allowed) considered as intra- or intergenomic homologous regions. Gene family size changes, based on orthogroups, were detected using CAFE5 [29] with the output from MCScanX to identify expansion and contraction of gene numbers within gene families and plotted onto the phylogenetic tree. GO term enrichment analysis of the expanded gene and contracted gene families was conducted using the TopGO package in R on the omicsbox GO annotations. Testing was performed following the topGO vignette and GO terms with low observed or expected counts ($< 5$) were removed to avoid false positives.

The genes of our assembly and the published assembly (GWHBCJH00000000) were compared and grouped by performing two protein blasts (blastp -evalue 1e-06), one with each assembly as the reference, and a using CD-HIT v4.8.1 [30] with a similarity threshold of 95%. Groups from CD_HIT were analysed to identify singleton genes from each assembly and cases where multiple genes from one assembly grouped to a single gene from the other assembly.

## Population analysis

Leaf samples of 29 A. *rumphiana* trees were collected from three sites in Thailand: Prachuap Khiri Khan, Chumphon, and Surat Thani. A reduced representation library was prepared from genomic DNA for each sample following the MGIEasy DNA Library Prep Kit Instruction Manual (MGI Tech, Shenzhen, China) using the restriction enzyme TaqI. The resulting reads were mapped to our A. rumphiana genome assembly using minimap2 v2.24 [31] with default parameters and variants were called using GATK v4.1 [32] with default parameters. The variants were then filtered to remove calls with less than 10 reads and to select SNPs where at least 20 of the 29 samples received a call. A principal components analysis was performed for the SNPs where all samples received a call using the 'prcomp' function in R and plotted using the R package ggplot2. The polymorphic information content of each SNP was calculated as a dominant marker type [33]. Population structure was calculated using the program STRUCTURE [34] with settings of no missing, minimum read depth of 10, and minor

allele frequency set to 1%. The optimum k-value was determined from a k-value analysis using numbers from one to ten using 20 replicates per k-value.

## Results and discussion

### Genome assembly, scaffolding and comparison

The genome assembled into 24,094 Supernova-scaffolds totalling 499.6 Mb, the largest Supernova-scaffold was 2.74 Mb and the N50 was 294 kb, from 110.9 Gb of paired-end sequence data. Merqury analysis showed that the genome assembly was 95% complete with optimum k-mer size 19.43 and had a k-mer multiplicity of one (S1 Fig). Our assembly was similar to the published *A. rumphiana* genome (GWHBCJH00000000), which was 470.6 Mb in 304 contigs with the longest contig at 21.1 Mb, it was assembled using PacBio, but not scaffolded into chromosomes [8]. To investigate the size difference between assemblies we mapped our assembly against the published assembly (GWHBCJH00000000) and found that 91.2% of our assembly mapped to the published assembly (GWHBCJH00000000) with an average identity of 99.89%, consistent with these being two individuals of the same species. However, there were 17,727 Supernova-scaffolds, totalling 41.16 Mb, that did not map to the published genome assembly (GWHBCJH00000000) and 16 Mb (3.4%) of the published genome (GWHBCJH00000000) had no sequence that mapped to it. Both our assembly and the published assembly (GWHBCJH00000000) contained mitochondrial and chloroplast sequence, so the disparity is not a result of one assembly simply lacking these sequences. Mapping our raw reads to each assembly showed a 95.7% maprate to our assembly and a 93.3% maprate to the published assembly (GWHBCJH00000000). The Supernova-scaffolds were blasted against the NBCI nt database to exclude sequences that represented sample contamination and, therefore, represent novel presence/absence allelic variation. The blast results showed no major hits to any known sequence, however, a small percentage of the internal sequence of some large Supernova-scaffolds matched bacterial sequence at an average identity of 70-80%, suggesting the sequence integrated with the *A. rumphiana* genome at some point in the past.

We mapped the Supernova-scaffolds of our *A. rumphiana* assembly against the *A. marina* genome to identify sequence similarity. Approximately 68.7% of the *A. rumphiana* genome sequence can be aligned to *A. marina* sequence, covering 72% of the *A. marina* genome, with 87.7% average nucleotide identity. Accounting for unmapped sequence gives an overall genome similarity of 60.3% at the whole genome nucleotide level. To add some perspective to this statistic, we have previously applied the same whole-genome-mapping approach to two species of Brugiera that can produce a fertile hybrid offspring and found that approximately 73% of the sequence of one can map to the other [35]. In addition, we have mapped a genome assembly of a Thai Sonneratia ovata tree to the genome assembly of a Chinese Sonneratia ovata tree and found that 92% of one can map to the other [36]. This high similarity at the whole-genome nucleotide level shows how closely related these two species are, which is not surprising considering they occupy the same environment niche and were originally identified as the same species.

The genome size of *A. rumphiana* is similar to that of *A. marina*, which is 457 Mb reported by Natarajan et al. [5], 456 Mb reported by Friis et al 2021 or 480 Mb reported by Ma et al. [7], so reference based scaffolding of the genome assembly was investigated. The published *A. rumphiana* assembly (GWHBCJH00000000) was less fragmented and had the largest contigs, so the entire genome assembly of this was mapped against *A. marina* using NUCmer and plotted. The result showed a high level of whole-seqence contiguity to *A. marina* (S2 Fig) with each contig mapping to a single *A. marina* chromosome. Six of the largest contigs

are completely contiguous to and cover a full *A. marina* chromosome, with the majority of chromosomes covered by 2–4 contigs. Some chromosomes show small inversions that occur towards the middle of the chromosome, which likely represent centromere sequence. All 32 *A. marina* chromosomes are covered. If the A. rumphiana genome was structurally different to A. marina we would see parts of each A. rumphiana (GWHBCJH00000000) contig mapping to multiple A. marina chromosomes, especially the contigs that represent entire A. marina chromosomes. This high sequence contiguity shows that *A. marina* was not only similar enough to *A. rumphiana* to allow for it to be used as a reference for reference based assembly, but that it was remarkably similar considering how long ago the two species diverged. Following from this finding we used *A. marina* as a reference to scaffold our assembly of *A. rumphiana* and obtained 32 chromosome-level RagTag-scaffolds totalling 447.3 Mb, leaving 52.3 Mb of unplaced sequence (Fig 1). The resulting RagTag-scaffolds were aligned back to *A. marina* and shown to have the same level of sequence contiguity as the published *A. rumphiana* genome (GWHBCJH00000000).

## Genome annotation and salt genes

The *A. rumphiana* genome was annotated and found to have 37,347 genes with 22,414 of those contained within RagTag-scaffolds (S1 File). Functional analysis of these genes identified 30,511 genes with a known annotation. The gene set was assessed for completeness using BUSCO and found to be 94.5% complete with 89.5% single copy and 5% duplicated. This is a larger number of genes compared to the published *A. rumphiana* genome (GWHBCJH00000000) or the *A. marina* genome, which had 31,499 and 31,477 genes, respectively. The larger number of genes between the published *A. rumphiana* genome (GWHBCJH00000000) and ours can be explained by a difference in gene annotation parameters and the larger size of our assembly. Protein blast of the two *A. rumphiana* annotations against each other identified 21,380 genes in common, and 21,269 of these received a functional annotation (S1 Table). Grouping the genes using CD_HIT identified 50,529 gene clusters with 14,867 gene clusters that contained at least one gene from each assembly. Of the 14,867 gene clusters, there were 13,558 clusters that contained a single gene from each assembly and 1309 clusters that contained multiple genes. The multi-gene clusters contained 2576 genes from our assembly and 2086 genes from the published assembly and there were multiple cases where several smaller genes from our assembly grouped to a much longer gene from the published assembly, which partly explains why our assembly contained more genes. There were 19,388 genes from our assembly and 15,347 genes from the published assembly that formed single gene clusters. The majority of the genes from our assembly that did not receive a functional annotation (6164 of the 6536 genes) were in this group of single gene clusters.

Comparison of the genes showed that 22,554 *A. rumphiana* genes were homologous to 22,029 *A. marina* genes, which leaves a large number of genes that are too different between the species to be identified as homologous. This may make it more simple to identify salt genes that both species have in common, but makes it more likely that the exact mechanism of salt tolerance is at least slightly different between *A. rumphiana* and *A. marina*. The homologous gene set was searched for salt genes listed in Natarajan et al. [5] to identify salt genes that both species had in common. The list of *A. marina* salt genes contained 614 genes that were identified as playing some kind of role in salt tolerance and there were 671 *A. rumphiana* genes that had a best-hit blast match to 467 of these genes. This left 205 *A. marina* genes that did not have a best-hit *A. rumphiana* match, however a blast of *A. marina* against *A. rumphiana* shows that only 98 of these salt genes truly had no *A. rumphiana* match. Considering how much convergent evolution has occurred in mangrove species, this does not necessarily mean that the most

important salt tolerance genes are included in the list of homologous genes. Nevertheless, it is interesting to identify which known salt genes are most similar between the two species and which, if any, have experienced gene duplication. Not only was there a large difference between the number of salt genes that were similar enough to be identified as a homologue, but the mean percentage similarity of the salt gene homologues was lower (86.8%) than the mean percent similarity of the entire gene set (91.1%), showing that the salt genes are more diverse than would be expected based on the entire gene set (p-val 4.12e–11). This is particularly surprising considering the high similarity in salt tolerance mechanisms and recent divergence time.

The gene families that form the salt secreting glands on the leaves were identified in the *A. marina* genome as involving ion channels, aquaporins, plasmodesmata, and vesicles [5]. Nine genes or gene families were identified in the *A. marina* genome as contributing to the salt secreting gland. To identify homologues of these genes, we performed a blast search using the *A. marina* gene and identified homologues based on sequence similarity. There were 11 genes identified as sodium/hydrogen exchanger or cation/proton exchanger, 18 genes identified as cation/H(+) antiporter, and a single gene identified as a cation-chloride cotransporter. Five genes were identified as S-type anion channel, with 19 genes identified as a cyclic nucleotide-gated ion channel, and 17 genes annotated as a glutamate receptor. There were 28 genes identified as an aquaporin, 7 genes identified as a chloride channel, and 5 genes identified as sodium transporter HKT1. How much of a role each of these genes play, if any, in salt excretion remains to be elucidated.

## Phylogenetic tree

A phyogenetic tree of 76 mangrove and mangrove related species was recently published [8]. Here we generated a phylogenetic tree, using single copy orthologues, of our A. rumphiana assembly compared to some mangroves that were newly sequenced since the previous large phyogenetic tree, plus a selection of previous samples to aid comparison (Fig 2). Mangroves are a prominent example of convergent evolution, so comparing the gene content of various mangrove species may give some insight into salt tolerance evolution. The results suggested that *A. rumphiana* and *A. marina* separated from each other 17.8 million years ago (MYA), consistent with the previous finding. The gene orthologue set was used to estimate gene expansion or contraction of gene numbers within gene families using a birth and death process to model gene gain and loss across this phylogenetic tree (Fig 2). *A. rumphiana* gained genes in 45 gene families and lost genes in 357 gene families, while *A. marina* gained genes in 265 gene families and lost genes in 59. GO term enrichment analysis of the expanded gene families showed enrichment for protein serine/threonine kinase activity and DNA-binding transcription factor activity as well as several underrepresented GO terms (S2 Table). While contracted gene familied were enriched for DNA binding, protein binding, mRNA binding, and also protein serine/threonine kinase activity. While the over- and under-represented GO terms are biologically important, they occur in too many different pathways to offer any specific insight into the evolution of this species.

## Repeat sequence analysis

The total interspersed repeats accounted for 46% of the genome sequence and the most common known repeat type was Ty1/Copia and Gypsy/DIRS1 LTR elements (Table 1). The majority of repeats were in the category 'unclassified' and account for approximate 25.6% of the genome followed by LTR elements. Approximately 3% of the genome consists of DNA transposons, with the most common identified sequence being hobo-Activator and Tourist/Harbinger transposons.

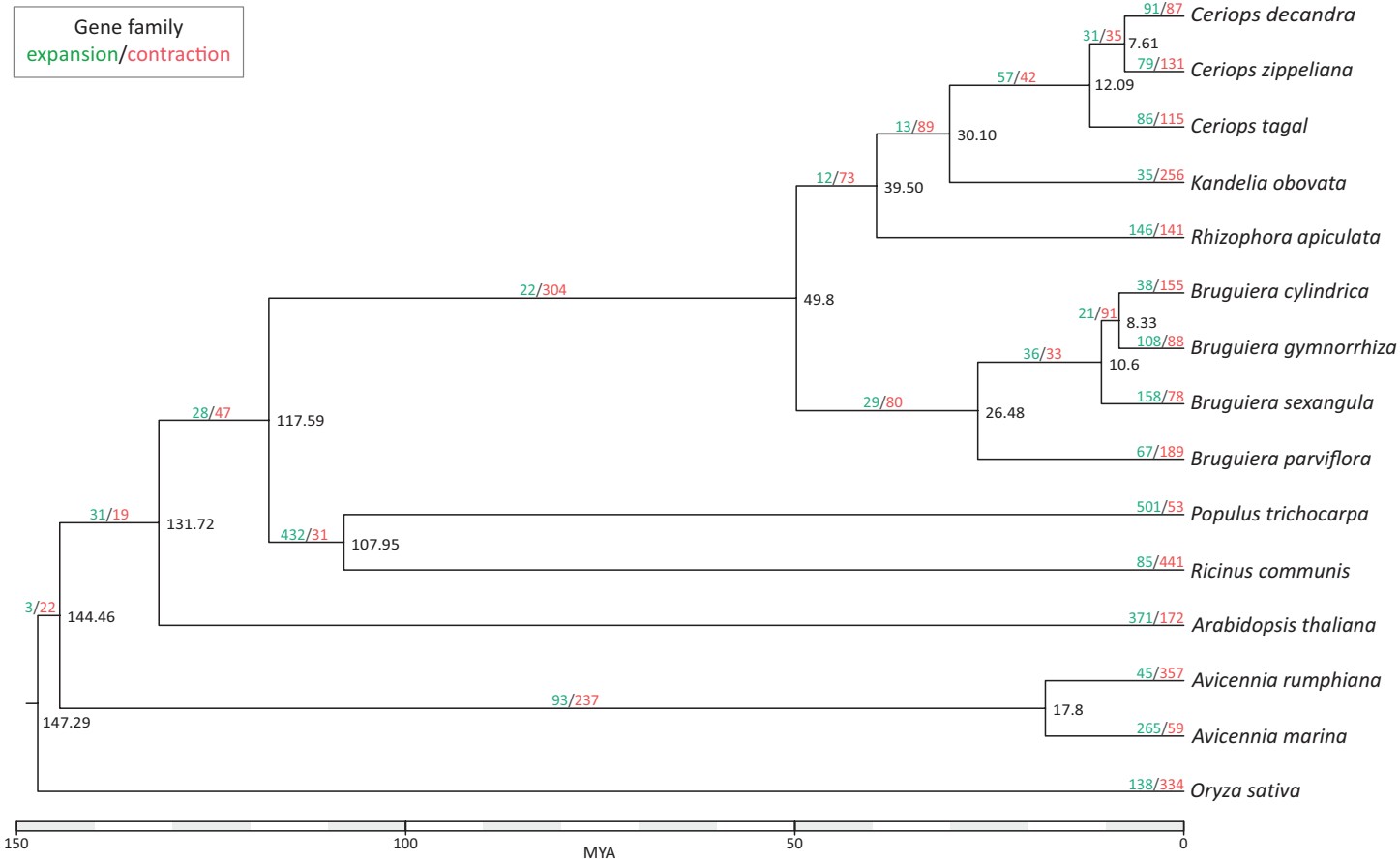

**Fig 2. Maximum-likelihood phylogenetic tree of Avicennia rumphiana.** Divergence times (million years ago) are indicated at each node. The number of gene families that have gained or lost genes are indicated as blue and red numbers, respectively.

## Population analysis

Proper management of mangrove species, which are being threatened by rising sea levels and deforestation, can be aided by measuring the genetic diversity of each species in conservation sites. A population with a low level of genetic diversity can be at increased risk of collapse from mechanisms such as inbreeding depression and reduced adaptability to biotic or abiotic stresses. Genetic diversity of A. rumphiana in Thai mangrove conservation sites were estimated by SNP variation present in trees sampled from multiple sites in Thailand. The total filtered SNP set contained 7,382 SNPs where all 29 samples were called. The polymorphic information content (PIC) ranged from 0.034 to 0.5 with the median PIC being 0.44, indicating high genetic diversity. A principle components analysis (PCA) showed a loose clustering effect of the samples according to the geographic location from which they were sampled (Fig 3), whereas STRUCTURE analysis showed two mixing populations with Chumpon and Surat Thani forming a single group (S3 Fig). Increasing the K-value in STRUCTURE added separation to the Prachuap Khiri Khan samples rather than the Chumpon and Surat Thani samples. The samples were collected from the east coast of three bordering Thai provinces in the Malay peninsula: Prachuap Kiri Khan in the north, Chumpon, and Surat Thani in the south. There is some overlap between samples from Prachuap Kiri Khan with those from Chumpon, consistent with trees coming from neighboring locations, but otherwise the samples show separation

**Table 1.  Repeat sequence statistics for the assembly of *Avicennia rumphiana*.**

| Total length | 499583238 bp | | |
|---|---|---|---|
| GC level: | 35.24% | | |
| Bases masked: | 236170459 bp (47.27%) | | |
| **Repeat** | **Number of elements** | **Length (bp)** | **Percentage** |
| SINEs: | 335 | 65174 | 17.47 |
| Penelope | 5 | 441 | 0 |
| LINEs: | 5638 | 2702032 | 0.54 |
| R2/R4/NeSL | 17 | 990 | 0 |
| RTE/Bov-B | 27 | 5312 | 0 |
| L1/CIN4 | 5541 | 2692078 | 0.54 |
| LTR elements: | 89144 | 84521762 | 16.92 |
| BEL/Pao | 23 | 1839 | 0 |
| Ty1/Copia | 44580 | 40612852 | 8.13 |
| Gypsy/DIRS1 | 39653 | 40767258 | 8.16 |
| Retroviral | 1787 | 515299 | 0.1 |
| DNA transposons: | 21706 | 14889038 | 2.98 |
| hobo-Activator | 3043 | 2452132 | 0.49 |
| Tc1-IS630-Pogo | 400 | 112862 | 0.02 |
| PiggyBac | 16 | 901 | 0 |
| Tourist/Harbinger | 2196 | 1146729 | 0.23 |
| Unclassified: | 398282 | 127841791 | 25.59 |
| Total interspersed repeats: | | 230019797 | 46.04 |
| Small RNA: | 2082 | 596603 | 0.12 |
| Satellites: | 65 | 3500 | 0 |
| Simple repeats: | 114002 | 4290570 | 0.86 |
| Low complexity: | 18944 | 913599 | 0.18 |

Indented repeat types are subtypes of the unindented type above.

based on location. The PIC values and PCA results showing sample separation show that a healthy level of genetic variation exists in the population. These findings support the IUCN rating of 'vulnerable' for this species, which is the least severe status in the 'threatened' category.

## Conclusion

The genome of *Avicennia rumphiana* was assembled and annotated. The 499.6 Mb genome was similar enough to *Avicennia marina* to allow reference based scaffolding, which produced 32 chromosomes containing 447.3 Mb of sequence. Annotation identified 37,347 genes, with 30,511 of those receiving functional annotation and 22,414 contained within chromosome scaffolds. Comparison of our A. rumphiana assembly to the published A. rumphiana assembly (GWHBCJH00000000) showed that 91.2% of our assembly mapped to 96.6% of the published assembly with an average identity of 99.89%, consistent with the two assemblies being from different individuals of the same species. Despite the high similarity in sequence, the annotated genes showed a lot of divergence with 21,380 common genes, while the remaining genes were different enough to not match under stringent settings. These differences are likely a combination of real differences between the two individual trees that were sequenced and differences in the gene annotation algorithms used between the two assemblies. Similar

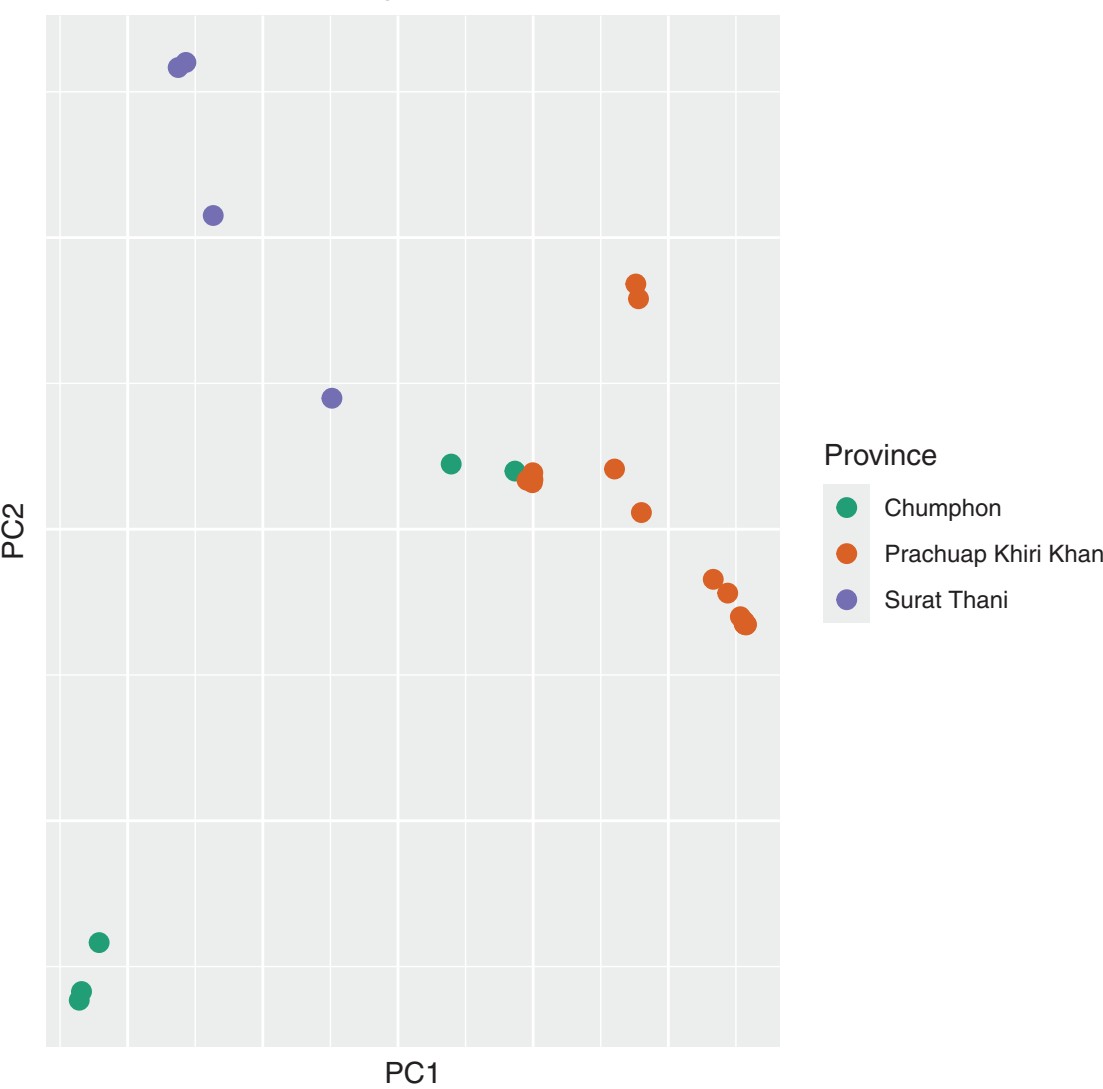

**Fig 3. Principal component analysis of 29 *Avicennia rumphiana* trees from Thailand.** Individuals are color coded according to location on the east coast of the Malay Peninsula for the three bordering provinces of Prachuap Kjiri Khan (north), Chumpon, and Surat Thani (south) in Thailand. The sum of variance explained by PC1 and PC2 is 66.25%.

comparison to *A. marina* showed that approximately 68.7% of *A. rumphiana* aligns to 72% of the *A. marina* genome with an average nucleotide identity of 87.7%. Gene comparison identified 20,554 common genes between *A. rumphiana* and *A. marina*. The comparative genetics conducted in this study is good evidence that *A. rumphiana* is not a variety of *A. marina* and the naming of the species should reflect this.

## Supporting information

**S1 Fig. Merqury spectra-asm.st plot showing the k-mer distribution of the Supernova-scaffolds and the read-only distribution.** A single well-defined peak indicates a complete assembly with minimal redundancy.
(PNG)

**S2 Fig. Whole-genome sequence of the published *A. rumphiana* assembly contigs (y-axis) mapped against whole-genome sequence of *A. marina* chromosomes (x-axis).** Cases where multiple A. rumphiana contigs map to a single A. marina chromosome appear as stacked boxes where the length of the y-axis is proportional to the amount of sequence that maps, resulting in small regions appearring as individual lines. Blue dots indicate direct sequence alignment, while red dots indicate inverted sequence alignment.
(PDF)

**S3 Fig. STRUCTURE analysis (K = 2) of 29 A. rumphiana samples from three bordering provinces in Thailand: Prachuap Khiri Khan (PKN), Chumpon (CMP), and Surat Thani (SNI).**
(TIF)

**S1 Table. Protein blast results of matching genes between our *A. rumphiana* assembly and the published *A. rumphiana* assembly including functional annotation.**
(XLSX)

**S2 Table. GO term enrichment analysis results of the expanded and contracted gene family GO terms.**
(XLSX)

**S1 File. Gene annotation data (GFF) of *A. rumphiana* assembly.**
(7Z)

## Aknowledgments

The authors would like to acknowledge the Ranong Biosphere Reserve, which is a UNESCO recognized reserve to help maintain mangrove diversity.

## Author contributions

**Conceptualization:** Jeremy Ross Shearman, Wirulda Pootakham, Sithichoke Tangphatsornruang.

**Data curation:** Jeremy Ross Shearman, Chaiwat Naktang, Chutima Sonthirod, Wasitthee Kongkachana.

**Formal analysis:** Jeremy Ross Shearman, Chaiwat Naktang, Chutima Sonthirod, Wasitthee Kongkachana.

**Funding acquisition:** Sithichoke Tangphatsornruang.

**Investigation:** Jeremy Ross Shearman, Sonicha U-thoomporn, Nukoon Jomchai.

**Methodology:** Jeremy Ross Shearman.

**Resources:** Chatree Maknual, Suchart Yamprasai, Poonsri Wanthongchai.

**Supervision:** Wirulda Pootakham, Sithichoke Tangphatsornruang.

**Visualization:** Jeremy Ross Shearman.

**Writing – original draft:** Jeremy Ross Shearman.

**Writing – review & editing:** Jeremy Ross Shearman, Wirulda Pootakham, Sithichoke Tangphatsornruang.

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
