## [Decision Letter · Decision Letter 0]

15 Sep 2024

PONE-D-24-21910Assembly of the salt-secreting mangrove Avicenia rumphianaPLOS ONE

Dear Dr. Shearman,

Thank you for submitting your manuscript to PLOS ONE. After careful consideration, we feel that it has merit but does not fully meet PLOS ONE’s publication criteria as it currently stands. Therefore, we invite you to submit a revised version of the manuscript that addresses the points raised during the review process.

Dear Authors,

The reviewers have submitted their comments and recommendations. Please revise the manuscript accordingly and provide responses to all reviewer comments.

We look forward to receiving your revised manuscript.

Kind regards,

Phuping Sucharitakul

Academic Editor

PLOS ONE

“This work was funded by the National Science and Technology Development Agency, Thailand.”

5. We are unable to open your Supporting Information file [Supplementary File 1 Arumphiana.gff3.7z]. Please kindly revise as necessary and re-upload.

Reviewers' comments:

Reviewer's Responses to Questions

**Comments to the Author**

1. Is the manuscript technically sound, and do the data support the conclusions?

Reviewer #1: No

2. Has the statistical analysis been performed appropriately and rigorously? 

Reviewer #1: No

3. Have the authors made all data underlying the findings in their manuscript fully available?

Reviewer #1: No

4. Is the manuscript presented in an intelligible fashion and written in standard English?

Reviewer #1: Yes

5. Review Comments to the Author

Reviewer #1: This manuscript describes a highly fragmented draft genome of the salt-secreting mangrove Avicennia rumphiana to supplement the existing contig-level “high quality” RefSeq assembly. The rationale provided for the additional genome was the possibility of missing sequences/genes and expanding coverage of the A. rumphiana pan-genome. The new assembly was created with 10x Genomics linked reads and scaffolded to chromosome-level using a different species in the genus.

CAFE analysis was used to identify expansions or contractions of gene families, but with an unclear rationale for the other genomes used. The authors also appear to add some population genomics data for 29 trees, although the rationale for this is not explained in the introduction and the analysis is very shallow.

Overall, there appears to be some useful data in this paper. However, in its current form, it is let down by insufficient clarity of methods, a lack of clear rationale behind experimental design, and a confusing presentation of results. As a consequence, most of the conclusions are not currently supported by the data presented. The major exception is the final line, that A. rumphiana is not a variety of A. marina and the naming of the species should reflect this. (See implications of this in point A1.) However, it is not clear that this is a new result (see cited ref 8).

A. MAJOR COMMENTS/REVISIONS

A1. L21-23: “Sequence comparison showed that 68.7% of the A. rumphiana genome aligned to A. marina sequence, covering 72% of the A. marina genome at an average nucleotide identity of 87.7%, suggesting A. marina is suitable for reference based scaffolding of the A. rumphiana contigs.” This does not seem close-enough to me for reference-based scaffolding. I would like to see some justification for this claim. (The authors conclude over 17 million years since a common ancestor!) In Results L204-208, it is not clear which contigs are being mapped and reported - the Methods indicate both assemblies were used. What was the agreement between the assemblies in terms of (a) coverage, and (b) synteny? I would like to see evidence from synteny-mapping of the new RagTag-scaffolded 10x draft genome against the high-quality PacBio contigs (e.g. RagTag scaffold off the PacBio contigs). If RagTag consistently scaffolds these regions off both references, it would increase confidence in the extra scaffolding off R. marina. Figure 1 is a useful supplementary figure, but not that informative as it reflects the low contiguity of the draft genome (see point A2). The interesting figure would be the comparison of the two A. rumphiana genomes. Given the high continguity of the reference reported on L214-220, a much more sensible approach would be to RagTag-scaffold the PacBio genome off R. marina and then RagTag the new fragmented draft genome off the inferred chromosome-level A. rumphiana genome. I would like to see whether this scaffolding strategy produced better synteny, completeness and gene models before accepting the more distant reference for scaffolding.

A2. Supernova produces scaffolds. It is therefore unclear to simply refer to “A. rumphiana scaffolds of our assembly” in the Methods, as this could be before or after scaffolding. The methods need to to be updated to be more precise and clear. For example, identying sytenic blocks of collinear genes is only really of relevance before RagTag scaffolding - otherwise, you are just reporting the effectiveness of the scaffolding. (Ironically, the more fragmented the assembly, the more syntenic but less reliable the result will be.) I think the authors may be incorrectly calling these scaffolds contigs, but the genome is not available to check. Please release the genome, and check the statistics are accurate.

A3.The CAFE5 analysis appears to use their annotation and the public annotation of other species. Expansions/contractions could therefore be annotation strategy differences rather than biological differences. Was anything done to test/control for this? (E.g. confirm results with an independent consistent reannotation of all genomes using a tool like GeMoMa.) The choice of reference genomes for this analysis was odd. Why include so many distant relatives? Why not use a more appropriate set from reference 8? It is also important to put the results of the phylogenomic analysis in the context of reference 8.

A4. The authors appear to identify up to 9% structural differences between the two individuals (L189-197). However, this could just represent incomplete assemblies. These results need to be supported and confirmed by (1) Merqury assessments of completeness of each genome, and (2) reciprocal read mapping using the raw sequencing reads from each assembly. Ideally, if possible, the RRS data would also be mapped onto each genome and the proportions failing to map to each would be reported. (Which reference was used for designing the RRS sequencing?)

A5. Whilst the larger number of genes (page 11) could be due to the larger assembly, it could also represent a lot of fragmented genes that inflate gene numbers. The authors should do some analysis of protein lengths and gene structure (e.g. see https://academic.oup.com/gigascience/article/7/9/giy095/5067871). Until this is done, I cannot agree with the conclusion (L358-360): “These differences are likely a combination of real differences between the two individual trees that were sequenced and differences in the gene annotation algorithms used between the two assemblies.” Assembly quality remains the most likely explanation for much of the difference. Annotation quality differences could have a big impact on CAFE5 analysis (see A3). As with elsewhere, it is not always clear in the annotation discussion when the authors are referring to which A. rumphiana assembly. Please give the assemblies clear names and version numbers to enable specific descriptions of results.

A6. The popgen analysis is lacklustre and incomplete. What percentage of variation was explained by the PCA? Was there any population structure supporting different populations or lack of gene flow? (e.g. STRUCTURE) The Chumpon samples show quite different clustering. Why? Are two individuals hybrids? More explanation is needed as to how these results “shows that a healthy level of genetic variation exists in the population” (L335).

B. MINOR COMMENTS

B1. The genus is spelt incorrectly in the paper title.

B2. What are the colours in Fig 1?

B3. Can Fig 3 be made colour-blind friendly, please?

6. PLOS authors have the option to publish the peer review history of their article (what does this mean? ). If published, this will include your full peer review and any attached files.

**Do you want your identity to be public for this peer review?** For information about this choice, including consent withdrawal, please see our Privacy Policy .

Reviewer #1: **Yes: ** Richard J Edwards

---

## [Author Response · Author response to Decision Letter 0]

21 Oct 2024

All numbers referring to line changes are to the track changes copy.

Reviewer #1: This manuscript describes a highly fragmented draft genome of the salt-secreting mangrove Avicennia rumphiana to supplement the existing contig-level “high quality” RefSeq assembly. The rationale provided for the additional genome was the possibility of missing sequences/genes and expanding coverage of the A. rumphiana pan-genome. The new assembly was created with 10x Genomics linked reads and scaffolded to chromosome-level using a different species in the genus.

Response: Yes, unfortunately while we were still assembling the genome another group published theirs first. Their manuscript, however, focused on a broad-scale comparison of >70 different species without any specific insight into to any one species.

Reviewer #1: CAFE analysis was used to identify expansions or contractions of gene families, but with an unclear rationale for the other genomes used. The authors also appear to add some population genomics data for 29 trees, although the rationale for this is not explained in the introduction and the analysis is very shallow.

Response: The first sentence was replaced with “A phyogenetic tree of 76 mangrove and mangrove related species was recently published [8]. Here we generated a phylogenetic tree, using single copy orthologues, of our A. rumphiana assembly compared to some mangroves that were newly sequenced since the previous large phyogenetic tree, plus a selection of previous samples to aid placement” line 351-356

added the sentence: “Mangroves are a prominent example of convergent evolution, so comparing the gene content of various mangrove species may give some insight into salt tolerance evolution.” to line 356-358

added the sentence: “Proper management of mangrove species, which are being threatened by rising sea levels and deforestation can be aided by measuring the genetic diversity of each species in conservation sites. Genetic diversity of A. rumphiana in Thai mangrove conservation sites were estimated by SNP variation present in trees sampled from multiple sites in Thailand. ” to line 383-387

A part of the rationale was to add something extra after we were scooped, but we can’t really say that in the manuscript.

Reviewer #1: Overall, there appears to be some useful data in this paper. However, in its current form, it is let down by insufficient clarity of methods, a lack of clear rationale behind experimental design, and a confusing presentation of results. As a consequence, most of the conclusions are not currently supported by the data presented. The major exception is the final line, that A. rumphiana is not a variety of A. marina and the naming of the species should reflect this. (See implications of this in point A1.) However, it is not clear that this is a new result (see cited ref 8).

Response: We went through the materials and methods and tried to improve the clarity of exactly what was done and we attempted to extend this improved wording throughout the results, discussion, and conclusion. The final line you mentioned is a main focus of this manuscript.

When uploading the genome and data to NCBI they did not allow us to use any name other than ‘Avicennia marina var. rumphiana’, so while it is not necessarily a new result, it is an ongoing problem, and in the author’s opinion, one that is best corrected with a genome assembly and analysis, as presented here. Actually, when we began this project we were given the samples (by our collaborators at the Thai government Department of Marine and Coastal Resources, who work with the mangrove conservation efforts of Thailand) as ‘Avicennia lanata’ and there are many papers that refer to it as this species name, with no mention that it is also known as ‘Avicennia marina var. rumphiana’ or ‘Avicennia rumphiana’. Some literature mentions one of the synonym names, but rarely both. It is for this reason that we included all three names in key words. Since the prior publication of this species’ genome focused only on comparative genomics of a large number of species, this would be the first paper to present the genome and an analysis that focuses solely on this species.

Reviewer #1: A. MAJOR COMMENTS/REVISIONS

A1. L21-23: “Sequence comparison showed that 68.7% of the A. rumphiana genome aligned to A. marina sequence, covering 72% of the A. marina genome at an average nucleotide identity of 87.7%, suggesting A. marina is suitable for reference based scaffolding of the A. rumphiana contigs.” This does not seem close-enough to me for reference-based scaffolding. I would like to see some justification for this claim. (The authors conclude over 17 million years since a common ancestor!)

Response: L21-23 is just the abstract summary, the full evidence is in the results and discussion L263-278 where we mapped not only our A. rumphiana to the A. marina genome, but also the published Pacbio assembly. We found remarkably high whole-genome-sequence contiguity with several instances of a single A. rumphiana PacBio contig being fully contiguous with a single A. marina chromosome (see fig S1).

As far as I know, the comparative genomics method I use to compare whole-genome-sequences is not widely used, so while that number seems low relative to gene level comparisons, it actually is quite high for whole-genome-sequence comparisons. I have added some references of other instances that I applied this mapping approach L253-258. I also realized that it was not described in the materials and methods and this oversight has been corrected L155-169.

A. marina was the closest genome available, and remains the closest genome with a chromosome level assembly available. So even though the divergence is 17 million years, the two species still have very high sequence contiguity.

Reviewer #1: In Results L204-208, it is not clear which contigs are being mapped and reported - the Methods indicate both assemblies were used. What was the agreement between the assemblies in terms of (a) coverage, and (b) synteny?

Response: The wording was corrected to specify that we are mapping the Supernova-scaffolds of our assembly against the A. marina genome. The coverage statistics are listed in L248-251 “Approximately 68.7% of the A. rumphiana genome sequence can be aligned to A. marina sequence, covering 72% of the A. marina genome, with 87.7% average nucleotide identity”.

The authors did not specifically look at gene synteny and instead focused on whole-sequence alignment, which is incredibly more informative than simple gene order. Whole-genome-sequence alignment between the two species show nearly full contiguity as seen in Figure 1 (our A. rumphiana) and Figure S1 (published A. rumphiana).

Reviewer #1: I would like to see evidence from synteny-mapping of the new RagTag-scaffolded 10x draft genome against the high-quality PacBio contigs (e.g. RagTag scaffold off the PacBio contigs). If RagTag consistently scaffolds these regions off both references, it would increase confidence in the extra scaffolding off R. marina.

Response: Since both A. rumphiana assemblies consist of a large number of contigs it becomes messy to try and plot them against each other and results a figure that contains hundreds or thousands of plots. The authors considered that describing the results in L263-274 and showing the whole-genome-sequence mapping of the PacBio contigs in fig S1 would be more effective (considering that some contigs were an entire chromosome and had full sequence alignment) to convince the reader of the suitability of A. marina as a reference for reference based scaffolding as this limits the number of individual plots to the 32 chromosomes of A. marina. As mentioned above we looked at whole-sequence alignment, not gene synteny.

Reviewer #1: Figure 1 is a useful supplementary figure, but not that informative as it reflects the low contiguity of the draft genome (see point A2).

Response: Fig 1 is showing whole-genome-sequence mapping results, which shows a very high level of sequence contiguity. The red and blue scattered dots mostly represent repeat sequence that can map to multiple locations. The authors have attempted to remove this confusion and make the manuscript more clear by removing a remnant sentence (“The genome comparison figure was generated using Circos [21] based on blastp results of peptide sequences for A. rumphiana against A. marina.”) of a figure was not included, and by adding the figure numbers that were produced by plotting the whole-sequence alignment data (Figure 1 and Figure S1). In addition the section describing MCScanX was moved into the phylogenetic tree paragraph where its output was actually used, and the figure 1 legend was modified to better show that it is whole-genome sequence mapping results.

Reviewer #1:The interesting figure would be the comparison of the two A. rumphiana genomes. Given the high continguity of the reference reported on L214-220, a much more sensible approach would be to RagTag-scaffold the PacBio genome off R. marina and then RagTag the new fragmented draft genome off the inferred chromosome-level A. rumphiana genome. I would like to see whether this scaffolding strategy produced better synteny, completeness and gene models before accepting the more distant reference for scaffolding.

Response: Figure S1 shows whole-sequence alignment of the published pacBio contigs of A. rumphiana against A. marina, some of which are full chromosomes and show near perfect whole-sequence alignment. Perhaps it would be marginally better to first RagTag-scaffold the PacBio genome and then use that as the reference, however the data do not suggest it would be a big improvement and considering that it would require essentially a re-do of the whole project and replacement of all the database files, the authors consider that it is not necessary, especially since one could then argue that the additional RagTag step could be compounding any RagTag errors that might exist.

Reviewer #1: A2. Supernova produces scaffolds. It is therefore unclear to simply refer to “A. rumphiana scaffolds of our assembly” in the Methods, as this could be before or after scaffolding. The methods need to to be updated to be more precise and clear. For example, identying sytenic blocks of collinear genes is only really of relevance before RagTag scaffolding - otherwise, you are just reporting the effectiveness of the scaffolding. (Ironically, the more fragmented the assembly, the more syntenic but less reliable the result will be.) I think the authors may be incorrectly calling these scaffolds contigs, but the genome is not available to check. Please release the genome, and check the statistics are accurate.

Response: The authors were referring to the final output of Supernova as contigs in an attempt to avoid confusion with the scaffolds resulting from RagTag and concede that this just results in a different type of confusing writing. Instances of ‘contig’ relating to Supernova output were replaced with ‘Supernova-scaffold’ and instances of ‘scaffold’ replaced with ‘RagTag-scaffold’, while ‘contig’ for the PacBio assembly was left as ‘contig’. In addition, it was mentioned in the materials and methods that Supernova includes a scaffolding step so output will be referred to as ‘Supernova-scaffold’ L100-102. We have contacted NCBI to ensure that all data is released. See above about gene synteny.

Reviewer #1: A3.The CAFE5 analysis appears to use their annotation and the public annotation of other species. Expansions/contractions could therefore be annotation strategy differences rather than biological differences. Was anything done to test/control for this? (E.g. confirm results with an independent consistent reannotation of all genomes using a tool like GeMoMa.) The choice of reference genomes for this analysis was odd. Why include so many distant relatives? Why not use a more appropriate set from reference 8? It is also important to put the results of the phylogenomic analysis in the context of reference 8.

Response: The observed expansion/contractions are consistent with literature values making it unlikely that they would be largely influenced by differences in annotation, although this cannot be completely excluded. In addition, the annotation pipeline that we used is the current standard annotation pipeline, with the two most common being ‘EvidenceModeller’ and ‘Maker’, which both implement a merge of ab initio gene model prediction, RNAseq, and database protein sequences. The BUSCO results also support a high level of genome completeness.

The CAFE5 result is solely to add to the phylogenetic figure with the primary focus being the comparison to A. marina. The choice of species then is locked to the phylogenetic choice, which we previously attempted to defend as intended to supplement the phylogenetic tree from ref 8. Hopefully our attempt to justify the selection of species for the phylogeny thus extends to the CAFE5 result. However, if the reviewer considers this unacceptable we could simply remove the gene expansion/contraction numbers with no material change to the manuscript.

Reviewer #1: A4. The authors appear to identify up to 9% structural differences between the two individuals (L189-197). However, this could just represent incomplete assemblies. These results need to be supported and confirmed by (1) Merqury assessments of completeness of each genome, and (2) reciprocal read mapping using the raw sequencing reads from each assembly. Ideally, if possible, the RRS data would also be mapped onto each genome and the proportions failing to map to each would be reported. (Which reference was used for designing the RRS sequencing?)

Response: (1) We ran merqury L102-104 and reported on genome completeness statistics L223-225 showing that the assembly is high quality.

(2) We added a small sentence about the percent of our raw reads that could map to each assembly in the comparative genetics sections of the materials and methods L170-172 and the results and discussion L238-239. The raw reads of the PacBio assembly is not publicly available, so we could not map those.

The text was update to clearly show that we mapped the RRS to our genome assembly L207-208.

We didn’t map the RRS to the published assembly. The proportion of reads failing to map would be a mixture of failed sequencing reads, bacterial/viral contamination, errors in the reference, and real genetic variation. Since these are different trees collected from different locations each sample would be a different combination of those causes, so the mapping rates would not be particularly informative.

Reviewer #1: A5. Whilst the larger number of genes (page 11) could be due to the larger assembly, it could also represent a lot of fragmented genes that inflate gene numbers. The authors should do some analysis of protein lengths and gene structure (e.g. see https://academic.oup.com/gigascience/article/7/9/giy095/5067871). Until this is done, I cannot agree with the conclusion (L358-360): “These differences are likely a combination of real differences between the two individual trees that were sequenced and differences in the gene annotation algorithms used between the two assemblies.” Assembly quality remains the most likely explanation for much of the difference. Annotation quality differences could have a big impact on CAFE5 analysis (see A3). As with elsewhere, it is not always clear in the annotation discussion when the authors are referring to which A. rumphiana assembly. Please give the assemblies clear names and version numbers to enable specific descriptions of results.

Response: Our BUSCO results were 94.5%, which suggests a high quality assembly and annotation. The example paper is essentially the pipeline we used, except we used EvidenceModeller instead of Maker, but the individual steps are the same. As described in the introduction, evidence coming out in recent years of additional high-quality assemblies has shown that structural variants between individuals often contain genes, with some estimates that a single genome assembly may only capture 80-90% of the genes that exist in the species. Of course having said that, there is considerable variability in annotation methods/parameters where

---

## [Decision Letter · Decision Letter 1]

21 Nov 2024

PONE-D-24-21910R1Assembly of the salt-secreting mangrove Avicennia rumphianaPLOS ONE

Dear Dr. Shearman,

Thank you for submitting your manuscript to PLOS ONE. After careful consideration, we feel that it has merit but does not fully meet PLOS ONE’s publication criteria as it currently stands. Therefore, we invite you to submit a revised version of the manuscript that addresses the points raised during the review process.

We look forward to receiving your revised manuscript.

Kind regards,

Phuping Sucharitakul

Academic Editor

PLOS ONE

Journal Requirements:

Additional Editor Comments:

Dear Authors,

The reviewers have submitted their comments on your manuscript. Kindly address their feedback and revise the manuscript accordingly.

Reviewers' comments:

Reviewer's Responses to Questions

**Comments to the Author**

1. If the authors have adequately addressed your comments raised in a previous round of review and you feel that this manuscript is now acceptable for publication, you may indicate that here to bypass the “Comments to the Author” section, enter your conflict of interest statement in the “Confidential to Editor” section, and submit your "Accept" recommendation.

Reviewer #2: (No Response)

Reviewer #3: All comments have been addressed

2. Is the manuscript technically sound, and do the data support the conclusions?

Reviewer #2: Partly

Reviewer #3: Yes

3. Has the statistical analysis been performed appropriately and rigorously? 

Reviewer #2: No

Reviewer #3: Yes

4. Have the authors made all data underlying the findings in their manuscript fully available?

Reviewer #2: No

Reviewer #3: Yes

5. Is the manuscript presented in an intelligible fashion and written in standard English?

Reviewer #2: Yes

Reviewer #3: Yes

6. Review Comments to the Author

Reviewer #2: OK

Reviewer #3: 1) Just a comment regarding the setence (abstract):

“...Sequence comparison showed that 68.7% of the A. rumphiana genome aligned to A. marina sequence, covering 72% of the A. marina genome at an average nucleotide identity of 87.7%, suggesting A. marina is suitable for reference based scaffolding of the A. rumphiana contigs...”

*I believe authors should explain better this fact, in face of evolution features.

2) *for ecological meaning (population), I believe authors should explain better how health level exist, regarding of "...genetic variation exists in the population..." and the relationship with "...which is the least severe status in the ‘threatened' category...”

7. PLOS authors have the option to publish the peer review history of their article (what does this mean? ). If published, this will include your full peer review and any attached files.

**Do you want your identity to be public for this peer review?** For information about this choice, including consent withdrawal, please see our Privacy Policy .

Reviewer #2: No

Reviewer #3: No

---

## [Author Response · Author response to Decision Letter 1]

8 Jan 2025

Comments to the author:

This paper described a mangrove species A. rumphiana genome assembly and identification of salt genes. In general, there are some useful data generated in this paper. However, the mechanisms underlying needs to be explored. For example, we would like know which genes respond to salinity environment, which provides valuable insight into the molecular mechanisms of salinity tolerance in A. rumphiana. Some concerns needed to be addressed before further consideration.

1. L95-97: The sequenced data should be uploaded to public platform, such as NCBI or NGDC (https://ngdc.cncb.ac.cn/gsa). In this study, authors used short sequenced reads (150bp) to do de novo assembly of A. rumphiana genome. However, the repetitive genome regions are often difficult to assembly using short-read sequencing technologies. The published A. rumphiana assembly (GWHBCJH00000000) also revealed that there are more than 50% repetitive sequence in the whole genome. Therefore, it’s required to add new sequenced data, such as PacBio or ONT Long-read data, even Hic-data.

Response: The raw data should have been available at the time of review using the accession in the ‘Accession codes’ section at the end of the manuscript, upon checking, it is currently available at the end of the list of raw sequence data (SRX24708924).

10x genomics is not just short read sequencing, it uses emulsion PCR to perform barcoded sequencing of high molecular weight DNA (https://ucdavis-bioinformatics-training.github.io/2018-Dec-Genome-Assembly/10x-supernova/10x-supernova.html) to achieve a level of information similar to, or even in some cases, better than current long read technologies. While we would like to add Hi-C, we do not have the funding for it and must publish what we have. We think we have provided sufficient evidence to support use of reference based scaffolding instead.

2. L103-105: The authors present a chromosome-level scaffolds of A. rumphiana using a reference-guided approach. Why did you choose A. marina as reference genome for scaffolding? According to the phylogenetic tree from He et al. study (paper: Evolution of coastal forests based on a full set of mangrove genomes), Avicennia alba is closer mangrove species to A. rumphiana compared to A. marina. Additionally, several reference genomes of A. marina are publicly available, which A. marina genome is better for your genome assembly of A. rumphiana, why?

Response: We added the sentence “Considering only species with a chromosome-level genome assembly available, A. marina is the most closely related species to A. rumphiana.” to the introduction L63-65. A. alba was only assembled at the contig level.

We chose the assembly from Natarajan et al since it was done using long PacBio reads and had the annotation for salt genes available. However, there is no indication as to which, if any, of the 3 genome assemblies is better. We added the accession number of the A. marina assembly that was used L101.

3. L115-122: The genome annotation was not correct. To predict genes accurately in A. rumphiana assembly, the genome should be masked for repeats before genome annotation. This will avoid the prediction of false positive gene structures in repetitive and low complexity regions. In addition, the authors only predicted genes using ab initio prediction program Augustus. RNA-seq data from different tissues of A. rumphiana should be used to improve gene prediction accuracy.

Response: The annotation was performed on the masked genome, the wording has been corrected to reflect this L118. Annotation was performed using EvidenceModeller, which combined gene prediction and protein sequence from related species in the database. We do not have RNA-seq data due to budget constraints, but it would be slightly better if we did.

4. L186-L187: The expansion and contraction of gene families served as a key source for plants to acquire adaptive function. For this purpose, GO and KEGG enrichment analysis need to be performed.

Response: We performed GO analysis on the group of expanded genes and the group of contracted genes. KEGG pathway data was sparse for these two groups, so we decided that the results would not be reliable and focused only on GO analysis. This was added to the materials and methods, line 192-196, and discussed in lines 378-385.

5. L198-205: The sequenced data should be uploaded to public platform, such as NCBI or NGDC (https://ngdc.cncb.ac.cn/gsa). After sequencing, a quality control of raw data was required to perform and low-quality reads need to be removed. In addition, detailed parameters, filtering condition of software for reads alignment and variants calling are not clear.

Response: The data is available on NCBI (see comment 1). WGS read quality was checked and found to be good, but reads were not filtered prior to using the Supernova assembly software because of the barcoded nature of the reads. We added ‘with default parameters’ to the appropriate sections. Filtering was also not performed for the population samples as the mapping software excludes low quality alignments by default.

6. L227-228 and L243-244: The authors revealed that approximately 68.7% of the A. rumphiana genome sequence can be aligned to A. marina sequence, and 91.2% of A. rumphiana can be mapped to the published assembly (GWHBCJH00000000). This made me confused. Why did not you choose published assembly (GWHBCJH00000000) as your reference genome for scaffolding? Additionally, the assembly completeness of genome was need to be evaluated using BUSCO software and LTR Assembly Index (LAI) metric.

Response: The published assembly is not at the chromosome level, although we did show that some of their contigs were whole chromosomes. We did perform BUSCO L126-128 & L305-307. We performed Merqury analysis L103-105 & L227-229, which gives a reference free genome assembly quality analysis. Adding another seems excessive.

7. L383 and L290: The authors performed the population analysis and identification of salt genes, however, no further analysis was done for data mining, such as genetic diversity, genotype-environment association analysis, etc. The main problem is that it lacks innovation and new insight.

Response: The PCA, PIC and STRUCTURE analysis reports on genetic diversity to the full extent that the data allows. We did not collect and data that can be used to identify any kind of genotype-environment associations, that is outside the scope of this study.

Reviewer #3: 1) Just a comment regarding the sentence (abstract):

“...Sequence comparison showed that 68.7% of the A. rumphiana genome aligned to A. marina sequence, covering 72% of the A. marina genome at an average nucleotide identity of 87.7%, suggesting A. marina is suitable for reference based scaffolding of the A. rumphiana contigs...”

*I believe authors should explain better this fact, in face of evolution features.

Response: We added that this shows A. marina is closely related to A. rumphiana in the abstract. We added some discussion about this to the results and discussion section L261-264

2) *for ecological meaning (population), I believe authors should explain better how health level exist, regarding of "...genetic variation exists in the population..." and the relationship with "...which is the least severe status in the ‘threatened' category…”

Response: We added the sentence “A population with a low level of genetic diversity can be at increased risk of collapse from mechanisms such as inbreeding depression and reduced adaptability to biotic or abiotic stresses.” to L 405-407

---

## [Editor Report · Decision Letter 2]

10 Jan 2025

Assembly of the salt-secreting mangrove Avicennia rumphiana

PONE-D-24-21910R2

Dear Dr. Shearman,

We’re pleased to inform you that your manuscript has been judged scientifically suitable for publication and will be formally accepted for publication once it meets all outstanding technical requirements.

Kind regards,

Phuping Sucharitakul

Academic Editor

PLOS ONE
---

## [Editor Report · Acceptance letter]

PONE-D-24-21910R2

PLOS ONE

Dear Dr. Shearman,

I'm pleased to inform you that your manuscript has been deemed suitable for publication in PLOS ONE. Congratulations! Your manuscript is now being handed over to our production team.

Kind regards,

on behalf of

Dr. Phuping Sucharitakul

Academic Editor

PLOS ONE